# Towards Imitation Learning to Branch for MIP: A Hybrid Reinforcement Learning based Sample Augmentation Approach

**Changwen Zhang, Wenli Ouyang,**\***Hao Yuan, Liming Gong, Yong Sun**
AI Lab, Lenovo Research
`{zhangcw5,ouyangwl1,yuanhao4,gonglm3,sunyong4}@lenovo.com`

**Ziao Guo, Zhichen Dong, Junchi Yan**
Department of Computer Science and Engineering, Shanghai Jiao Tong University
`{ziao.guo,niconi19,yanjunchi}@sjtu.edu.cn`

## Abstract

Branch-and-bound (B&B) has long been favored for tackling complex Mixed Integer Programming (MIP) problems, where the choice of branching strategy plays a pivotal role. Recently, Imitation Learning (IL)-based policies have emerged as potent alternatives to traditional rule-based approaches. However, it is nontrivial to acquire high-quality training samples, and IL often converges to suboptimal variable choices for branching, restricting the overall performance. In response to these challenges, we propose a novel hybrid online and offline reinforcement learning (RL) approach to enhance the branching policy by cost-effective training sample augmentation. In the online phase, we train an online RL agent to dynamically decide the sample generation processes, drawing from either the learning-based policy or the expert policy. The objective is to strike a balance between exploration and exploitation of the sample generation process. In the offline phase, a value function is trained to fit each decision's cumulative reward and filter the samples with high cumulative returns. This dual-purpose function not only reduces training complexity but also enhances the quality of the samples. To assess the efficacy of our data augmentation mechanism, we conduct comprehensive evaluations across a range of MIP problems. The results consistently show that it excels in making superior branching decisions compared to state-of-the-art learning-based models and the open-source solver SCIP. Notably, it even often outperforms Gurobi.

## 1 Introduction

As a general formulation and long-standing challenge, Mixed Integer Programming (MIP) (Zhang et al., 2023) spreads wide applications ranging from manufacturing (Artigues et al., 2009) to route planning (Halim & Ismail, 2019). Many exact algorithms (Lomnicki, 1965) have been proposed, with common adoption of *branch and bound* (Brucker et al., 1994). It recursively divides the solution space into a search tree and calculates the relaxation and boundary to prune the subtrees proved not to contain the optimal solution. At each iteration, two important decisions need to be made, including node selection and variable selection, which determine the next node to evaluate and select the variables to partition the search space. In this paper, we will focus on variable selection.

**Challenges:** Variable selection decisions often rely on heuristics from domain experts (Achterberg et al., 2005b), like strong branching, active constraint (AC) (Patel & Chinneck, 2007), pseudo cost branching (PC) (Hendel, 2015), reliability pseudo cost branching (RB). However, the expert rule-based policies can often only find the locally best variable to branch on (Achterberg et al., 2005a). For example, for the most famous expert strong branching, it targets to select the variables that deliver the best one-step progress in the dual bound improvements, which has been proved not always the 'golden standard' (Dey et al., 2021; Qu et al., 2022a). So, Glankwamdee & Linderoth (2011) proposed

---

\*Correspondence author. The work was partly supported by NSFC (92370201, 62222607).

the lookahead branching, which considered more dual bound information from deeper levels to deal with the potential local search limitations of strong branching. While in general, these effective approaches were usually computationally expensive. In this respect, an increasing number of studies are resorting to a learning-based branching strategy (Balcan et al., 2018), in which imitation learning (IL) on strong branching (Hussein et al., 2017) has shown superiority. However, strong branching cannot produce completely reliable labels due to the dual degeneracy (Cao et al., 2022b; Gamrath et al., 2020b) in the LP solution, which may cause the product score of strong branching to combine the improvements of the two child nodes close to zero. In addition, IL-based branching approaches usually require a large amount of training data and GPU resources for training.

To tackle the aforementioned challenges, we propose an innovative hybrid approach that combines online and offline RL[1] techniques to enhance the training samples for branching policy learning. It involves two distinct phases. In the online phase, we leverage an online RL agent, acting as a dynamic collector. Its primary role is to guide decisions regarding the choice of sample generation processes. These processes draw from two potential sources: the learning-based policy and the expert policy. Contrasting with solely relying on expert knowledge, the samples could be generated by the learning-based policy with the hope of bolstering exploration in unfamiliar scenarios, thereby enhancing the performance of IL. The intuition is that learning-based policy (Gasse et al., 2019) can deliver a smaller B&B tree than strong branching in some instances, as shown in the Appendix A.6. Besides, Oh et al. (2018) has proved that past good experiences can add exploration for imitation learning and may find a potentially better policy than only imitating the expert policy. The offline phase involves training an offline agent. This offline agent is tasked with fitting the cumulative reward function for each decision made. It operates as a filter, sifting through the generated samples to identify those with superior cumulative returns. These high-quality samples are then earmarked for training the branching policy. By doing so, we not only expect to improve the performance of the branching policy but also minimize the overall training cost. **The highlights of this paper are:**

**1)** We propose an iterative collection and filtering framework that leverages a combination of online and offline RL techniques to enhance the training samples. The online RL agent serves as a collector responsible for determining the sample generation process, and choosing between the learning-based policy and the expert policy. The offline RL agent functions as a filter, identifying samples with high cumulative returns. This framework enhances the efficacy (inference performance) and efficiency (training complexity) of the learning-based branching methods. **2)** Extensive experiments on MIP show that it consistently outperforms the default heuristic policy adopted in open-source solver SCIP. Compared with learning-based algorithms, including the best-performing methods in the Machine Learning for Combinatorial Optimization (ML4CO) 2021 competition[2], our method can further enhance the branching policy, illustrating its effectiveness as a plugin orthogonal to peer methods.

**Difference to existing works:** Table 1 compares our approach with existing works in detail. The most critical difference is the hybrid sample collection and filtering scheme with online and offline RL agents. Peer works mainly depend on either the expert-based samples (Gasse et al., 2019; Cao et al., 2022a), or the pure self-generated samples with RL manner (Parsonson et al., 2023; Scavuzzo et al., 2022). To combine the best of the two worlds, Qu et al. (2022b) tries to mix the generated samples that lead to high performance with expert demonstration data within the training, purely by a predefined hyperparameter $G_0$. In this paper, we propose to train an online RL to adaptively and dynamically mix the best of two worlds, which is more reasonable and adaptive. In addition, we also proposed a new offline RL agent to further filter the mixed generated samples, minimizing the overall training costs and enhancing performance to some extent.

## 2 PRELIMINARIES AND RELATED WORK

**Preliminaries:** Mixed integer programming (MIP) can be defined with three elements: optimization objectives, decision variables, and constraints, which can be given by:

$$\arg\min_{\mathbf{x}}\{\ \mathbf{c}^\top\mathbf{x}|\mathbf{A}\mathbf{x} \leq \mathbf{b}, \mathbf{l} \leq \mathbf{x} \leq \mathbf{u}, \mathbf{x} \in \mathbb{Z}^q \times \mathbb{R}^{m-q}\} \tag{1}$$

where $\mathbf{x}$ is the decision variables of total size $m$. $n$ and $m$ denote the number of constraints and decision variables, respectively. $q$ denotes the number of integer variables, and the remaining $m - q$

---

[1]Offline RL is a filtration mechanism similar to (Chen et al., 2020).
[2]https://www.ecole.ai/2021/ml4co-competition

Table 1: Comparing our method with peer works.

| References | Sample Collection | | | Filtering | Iterative method | Training |
|---|---|---|---|---|---|---|
| | Expert based samples | Self-generated samples | How to mix | | | |
| Gasse et al. (2019) | Yes | No | / | / | No | Imitation |
| Cao et al. (2022a) | Yes | No | / | / | Yes | Imitation |
| Parsonson et al. (2023) | No | Yes | / | / | No | RL |
| Scavuzzo et al. (2022) | No | Yes | / | / | No | RL |
| Qu et al. (2022b) | Yes | Yes | Pre-defined distribution | / | No | RL |
| **HRL-Aug(Ours)** | Yes | Yes | Online agent | Offline agent | Yes | RL + Imitation |

variables are continuous. The objective is to minimize $\mathbf{c}^\top \mathbf{x}$ with the constraints $\mathbf{A}\mathbf{x} \leq \mathbf{b}$, where $\mathbf{A} \in \mathbb{R}^{n \times m}$ represents the constraint coefficient, and $\mathbf{b} \in \mathbb{R}^n$ denotes the right-hand-side vector. The $\mathbf{x}$ satisfying all the constraints and minimizing the objective is the optimal solution.

The flow of the B&B can be depicted as follows. Firstly, the raw MIP can be defined as the root node of a search tree. It recursively selects a node from the search tree by the node selection rule and then selects a variable to decompose the selected node. When branching on variable $x_i$, the optimal solution $x_i^*$ is first computed by imposing a linear programming relaxation, and the relaxed objective value can be defined as the dual bound. If $x_i^*$ does not meet the integrity constraint, the variable $x_i$ needs to be branched, decomposing the MIP from the current node into two sub-problems by adding two new constraints $x_i \geq \lceil x_i^* \rceil$ and $x_i \leq \lfloor x_i^* \rfloor$. The iterations continue until convergence or time up.

**Branching policy for branch and bound:** Expert rule-based branching policies are widely adopted in MIP solvers, including strong branching (Applegate et al., 1995), active constraint (Patel & Chinneck, 2007) etc, which are generally difficult to design, and can usually find suboptimal variables to branch on. Furthermore, some of them can be extremely time-consuming, like strong branching.

Learning-based policies were recently devised to replace the above expert rule-based policies, such as imitation learning on some best-performing expert rule-based policies, like strong branching (Applegate et al., 1995; Gasse et al., 2019; Alvarez et al., 2014; Gupta et al., 2020) and active constraint (Patel & Chinneck, 2007). Gasse et al. (2019) first proposed to utilize bipartite graph convolutional neural network (GCNN) to approximate strong branching decisions by IL. To make the expert samples closer to real-world applications and further improve the performance, Cao et al. (2022b) proposed a Dagger-like method to enhance the sample collection phase. These methods deliver better performance and efficiency. In addition, RL-based methods are also recently studied, which may produce even better branching policies compared with expert rule-based policies. Scavuzzo et al. (2022) proposed a new tree Markov Decision Process, a more suitable framework for learning to branch. Parsonson et al. (2023) proposed the retro-branching approach, by learning from deconstructing trajectories within the sub-trees to enhance the branching policy.

In general, purely exploiting what expert policies know limits the performance of imitation learning in some cases. While RL, free from expert rules, may be challenging in training, especially for some large-scale problems. Note the conducted extensive experiments on the RL-based methods in Appendix A.5, revealing the limitation of RL-based approaches. In this paper, we propose a hybrid approach that embeds RL into imitation learning to enhance the training samples, balancing between exploiting what experts know and exploring potential high-reward unknowns.

**Training set augmentation with RL.** Recently, RL-based data augmentation approaches have been widely studied. Wang et al. (2018) developed a monotonic advantage re-weighted IL strategy to enhance policy learning. Peng et al. (2019) proposed an advantage-weighted regression-based scheme for offline RL. Chen et al. (2020) proposed the Best-Action Imitation Learning (BAIL) approach, utilizing the V function to select actions for imitation learning that promised to be high-performing. With similar insights, Qu et al. (2022a) tends to utilize BAIL to improve the quality of expert samples for imitation learning. In general, these RL-based methods mainly rely on Monte Carlo return with a discount factor. However, in the branching scenario, the dual bound may remain unchanged for hundreds or even thousands of iterations, which means that the discount factor will interfere with the accurate evaluation of the samples. In this respect, we deal with the above issue by setting the discount factor as 1 to calculate the actual Monte Carlo return.

**Imitation learning and self-imitation learning:** Imitation learning (IL) has been widely adopted in machine learning for combinatorial optimization, including solving MIP (Zhang et al., 2023). As a special case for IL, the main idea of self-IL (Oh et al., 2018; Gangwani et al., 2018; Guo et al., 2019;

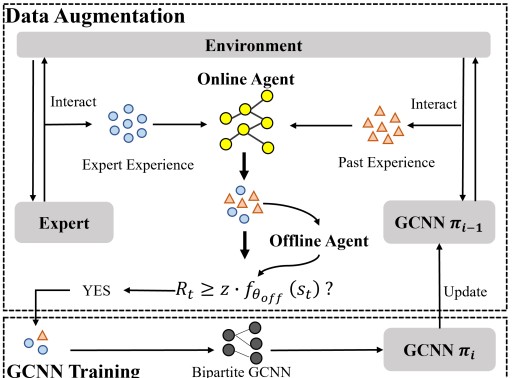 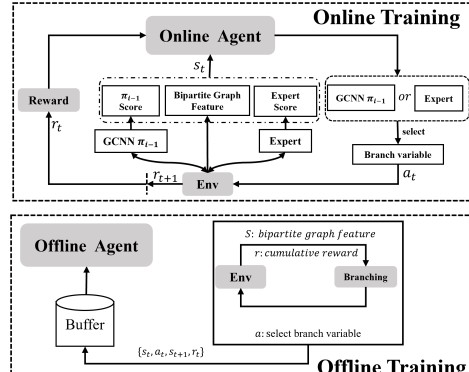

Figure 1: Overview of the proposed **HRL-Aug**. It is a new iterative approach hybrid with online and offline agents. An online agent is used to decide the sample generation processed from GCNN $\pi_{i-1}$ or experts to augment the training samples with higher cumulative rewards. An offline agent is devised to reduce the generated samples and further enhance the sample quality. The samples will be finally fed into GCNN iteratively for branching policy learning.

Zhu et al., 2014) is to learn the agent's past experiences to indirectly drive exploration. It is believed that the policy can be learned iteratively from the agent without any feedback from an external expert. It shows that past good experiences are helpful on hard exploration tasks and achieved good results in the Mujoco (Todorov et al., 2012). But in the branching scenario, each branching may have hundreds of thousands of candidate variables, which means there large state-actions space and it is a well-known difficulty for RL (Ecoffet et al., 2021), so we will use the online RL agent to adaptively choose samples generation processes, either from learning-based policy or the expert policy, with the aim of leveraging the strengths of both approaches.

## 3 METHODOLOGY

Fig. 1 depicts our approach with notations throughout this paper as listed in Table 2.

### 3.1 APPROACH OVERVIEW

Our approach, named **H**ybrid **R**einforcement **L**earning for Training Set **Aug**mentation (**HRL-Aug**), is a hybrid online and offline approach to enhance the training samples towards branching policy learning. The innovation lies in the introduction of the data augmentation scheme, as meticulously designed for the realm of imitation learning. It comprises three components: an online Reinforcement Learning (RL) agent, an offline RL agent, and an imitation learning-based policy, as shown in Fig 1. The hybrid online and offline data augmentation scheme generates high-quality samples for the IL-based GCNN network iteratively until the stop condition is met.

**The Online RL Agent** is utilized to select better samples from the expert policy or learning-based policy at each variable selection decision. It plays a pivotal role by facilitating profound exploration of uncharted scenarios. By striking a balance between exploration and exploitation, it effectively steer clear of local optimal solutions for branching variables.

**The Offline RL Agent** evaluates the cumulative rewards of generated samples from the online phase, and filtering the high-quality samples for subsequent policy training. As will be seen from our ablation study, it improves training efficiency.

**The Imitation learning-based policy** is trained with the generated samples from the hybrid online and offline agent. It can be regarded as a crucial validation of our data augmentation framework. It emerges

Table 2: Description of the notations in the paper.

| | |
|---|---|
| $s_t^I$ | state of offline reinforcement learning at node $t$ |
| $s_t^o$ | state of online reinforcement learning at node $t$ |
| $a_t^o$ | action of online reinforcement learning at node $t$ |
| $S_t^s$ | scores for branching variables given by expert at node $t$ |
| $S_t^p$ | scores for branching variables given by learned policies at node $t$ |
| $O_t$ | bipartite graph features at node $t$ |
| $R_t$ | cumulative reward at node $t$ |
| $r_t$ | reward at node $t$ |
| $\pi_i$ | policy trained by imitation learning at the $i$-th iteration |
| $A_t$ | candidate branching variable set at node $t$ |
| $z$ | hyper-parameter controlling the ratio of selected samples |
| $f_{\theta_{\text{off}}}$ | offline agent parameterized by $\theta_{\text{off}}$ |
| $f_{\theta_{\text{on}}}$ | online agent parameterized by $\theta_{\text{on}}$ |
| $\gamma$ | discount factor for Monte Carlo Return calculation |

as a pivotal component capable of supplanting expert-based policies within solvers, akin to SCIP as

exemplified in this study. GCNN is selected as the backbone. We take comprehensive tests on the generalization capabilities of our framework to other approaches, and the details are given in Sec. 4.4.

## 3.2 ONLINE RL PHASE

Imitation learning-based models have shown their superior performance over expert rule-based policies (Gasse et al., 2019). However, its performance largely depends on high-quality expert samples, which are difficult to collect. Recently, self-imitation learning has proved that using past good experiences to explore new behaviors may find a potentially better policy (Oh et al., 2018). However, the optimization may be completely aimless at the early training stage when starting from scratch, which may hurt the performance and greatly increase the complexity.

An intuitive idea is to combine the best of the two worlds: expert rules and past good policies, which requires an approach to decide the sample generation processes, denoted as the online phase in this paper. However, it is difficult to evaluate the branching decisions, making it a challenging task to choose samples from expert rules or learning-based policies. Moreover, in the branching scenario, samples are collected by iterative variable selection decisions, and the current decision largely depends on the former actions and states, making it a sequential decision problem. In this respect, we tend to define the online phase as a Markov Decision Process (MDP) and model the sample generation process with RL.

Firstly, we define the online RL phase as a sample selector and collector at each branching node. In this respect, the action $a_t^o$ was defined as the sample generation decision from expert rules or past learned policies at each node $t$, which is a discrete action space, only with two actions. The state $s_t^o$ is formulated as $s_t^o = (S_t^s, S_t^p, O_t)$, where $S_t^s$ denotes the expert score for variables, and the $S_t^p$ is the score from learned policies. $O_t$

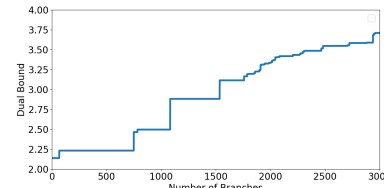

Figure 2: Example of the dual bound changes on a balanced item placement instance.

denotes the bipartite graph feature, and the details of features are listed in Table 11 in the Appendix. Secondly, it is essential to drive effective evaluation for samples. We utilize the dual bound changes as the reward $r_t$, and Fig. 2 depicts an example on an MIP instance. The dual bound may remain unchanged for hundreds of steps. This reveals that it may require thousands of or even longer steps to accurately evaluate the cumulative reward. However, with the discount factor $\gamma < 1$, the reward may have little contribution to the cumulative reward after thousands of branches, which can cause an inaccurate evaluation of the action. Hence we set $\gamma = 1$, and the cumulative reward $R_t$ at node $t$ is:

$$R_t = \sum_{v=t}^{t_{-1}} r_v \tag{2}$$

where $t_{-1}$ is the final expanded node in an episode. With the above definition, we can train the online RL agent with the advantage actor-critic method ($ActorCritic$). The agent dynamically decides sample generation processes, either from the expert rule or the GCNN policy $\pi_{i-1}$. In the training phase, GCNN policy $\pi_i$ is iteratively optimized and the online RL agent relies on this evolving GCNN to construct the input states. This introduces a potential challenge, as the previously trained online agent may become unreliable due to the evolving policy. To address this issue, we utilize the up-to-date $\pi_{i-1}$ to train the online agent. With the online update mechanism, we can optimize the online sample collection agent and obtain past good experiences, to achieve better exploration.

## 3.3 OFFLINE RL PHASE

With the online sample collection agent, we can easily obtain hybrid samples from both the expert rule and the GCNN policy. However, with the iterative sample generation process, the training samples constantly accumulate and require huge computing resources for imitation learning. In this respect, batch deep reinforcement learning (BDRL) (Fujimoto et al., 2018; Chen et al., 2020) was proposed to strive for both simplicity and performance by selecting high-quality samples with offline methods. In this paper, we leverage similar ideas and develop an offline RL agent to approximate the cumulative reward and reduce the data scale by filtering samples, thus reducing the training complexity.

For branching, the action $a_t^f$ is defined as the variable selection from the candidate branching variable set. The states for the offline learning phase at node $t$ can be defined as $s_t^f = (O_t, A_t)$, where $A_t$

denotes the candidate branching variable set, $O_t$ is the bipartite graph feature. The discount factor is still set as $\gamma = 1$, with the cumulative reward defined as Eq. 2, to evaluate the quality of the samples.

With the above definition, let offline agent denoted by a neural network characterized by $\theta_{\text{off}} = (w, b)$. $\theta_{\text{off}}$ take the $\{(O_t, A_t, R_t), t = 1, 2, ..., m\}$ collected from the online RL phase as input and output a real number to fit the cumulative reward, thereby obviating the need for exploration. Then the offline agent is regarded as a $\lambda$-regularized upper envelope for the following constrained optimization problem Eq. 3. This concept arises due to the inequality $f_{\theta_{\text{off}}}(O_t, A_t) \geq R_t$, and this prompts us to define $f_{\theta_{\text{off}}}(O_t, A_t)$ as the "upper envelope", a construct geared towards aligning the envelope as closely as feasible with $R_t$. Then the offline RL agent is used to select high-quality samples generated in the online phase. With the approximate cumulative reward generated from the offline agent, we can select the high-quality samples by Eq. 4, where $z$ is a hyper-parameter, controlling the ratio of the selected data samples, $f_{\theta_{\text{off}}}$ denotes the offline agent.

$$\min_{\theta_{\text{off}}} \sum_{t=1}^{m} [f_{\theta_{\text{off}}}(O_t, A_t) - R_t]^2 + \lambda \|\omega\|^2 \quad s.t. \ f_{\theta_{\text{off}}}(O_t, A_t) \geq R_t, \ t = 1, 2, \ldots, m \quad (3)$$

$$R_t \geq z f_{\theta_{\text{off}}}(O_t, A_t) \quad (4)$$

Note that the offline RL agent undergoes training solely during the first iteration, as empirical evidence suggests that its performance is already satisfactory at that point.

### 3.4 Hybrid Training pipeline

The hybrid online and offline RL agents are combined as a hybrid pipeline to further enhance the branching policy via training set augmentation, whose details are given in Algorithm 1. It operates iteratively to collect training samples, with each iteration involving the incremental training of the foundational GCNN model. For efficiency, the frequency of the online RL training will be controlled by the hyperparameter $Freq$ (see Line 6), and the offline RL agent is exclusively trained during the first iteration. In particular, the online agent will employ the most current GCNN policy for its training every time when the conditions in Line 6 are met. In the training phase for the online agent $f_{\theta_{\text{on}}}$, we will collect the bipartite graph features $O_t$, along with the scores $(S_t^s, S_t^p)$ from the expert rule and the GCNN policy $\pi_{i-1}$ as the states in Line 7. Afterward, the samples generated from the online phase (see Line 9-13) are used to train the offline agent $f_{\theta_{\text{off}}}$ at the first iteration in Line 15, which is capable of selecting samples with high cumulative rewards and will be applied for all the successive iterations. Finally, the generated samples are further aggregated to train the GCNN policy $\pi_i$, used for subsequent training. The pipeline persists until it reaches the iteration limit $N$.

## 4 Experiments

We compare competing learning-based approaches and SCIP's default branching rule. We run all the experiments on an Intel(R) Xeon(R) Silver 4210 2.20GHz CPU and an NVIDIA A100 GPU.

### 4.1 Experimental protocols

**Dataset.** We perform evaluation on popular binary integer programming problems: set covering, combinatorial auctions, maximum independent set, and capacitated facility location. Instance generation completely refers to (Gasse et al., 2019), and their variable and constraint sizes are listed in Table 3. For each problem, we generate 10,000 instances for training, 4,000 for validation, and 40 for testing. The test instances are all generated via different random seeds. The training and validation are performed on the Easy instances (see Table 3), and we also generated 40 Medium and Hard testing instances for each problem to further test the generalization ability. Besides, we conduct experiments on more difficult MIP problems from the ML4CO 2021 competition (ML4CO datasets) (Gasse et al., 2022), including Balanced Item Placement (1,083 columns, 195 rows), Workload Apportionment (61,000 columns, 64,314 rows), and Anonymous, which is from the MIRPLIB library (PAP, 2014) and inspired by real-world applications of large-scale problems.

**Hyperparameters.** All models are trained with Ecole (Prouvost et al., 2020). To verify the effect of different branching strategies, we disable SCIP's heuristic and default restart policy. In the training phase, we collect 10,000 training and 4,000 validation samples at each iteration, and the training iterations $N = 20, 50$ for binary integer programming problem and ML4CO datasets respectively, the frequency of training the online RL agent $Freq = 5$. At each iteration, the embedding size for the offline agent is 64, and the hidden layer size for both online and offline RL is set to 128. We set

the discount factor to 1, and use Adam optimizer with learning rate $3 \times 10^{-3}$ to train both RL agents. We use SCIP 7.0.3 as the backend solver, with a time limit of 1 hour for binary integer programming problems which are the same as (Gasse et al., 2019), and 900 seconds for ML4CO datasets.

**Evaluation.** As discussed above, for each binary integer programming problem, evaluation is performed on different difficulty levels (Easy, Medium, Hard), each with 40 instances using five different seeds, which amounts to a total of 200 solving attempts per method to report aggregated results over the same instance. We use similar metrics as those in (Gasse et al., 2019), including the 1-shifted geometric mean of the solving time to measure the solving efficiency (Time), the final node counts of the solved instances among all baselines (Nodes), and the number of times each branching policy delivers the fastest solving time, over the number of instances solved to optimal (Wins). Note that we also report the average per-instance standard deviation.

As it is too large to solve in a reasonable time, we evaluate methods with similar metrics as in ML4CO 2021 competition: average $DualGap$ and $dualbound$ within the solving time limit. $DualGap$ for each instance is:

$$DualGap(t) = \frac{|d(t)^* - p^*|}{max(|d(t)^*|, |p^*|)} \quad (5)$$

where $d(t)^*$ is the best dual bound at time $t$ and $p^*$ is the best known solution value for the instance. We also compare the cumulative reward within the fixed time bound by Eq. 6, where $Tc^\top x^*$ is an instance-specific constant w.r.t. the optimal objective $c^\top x^*$.

---

**Algorithm 1** Hybrid RL for the training set augmentation to enhance branching for B&B-based MIP.

---

**Input**: Training instances;
Number of samples generated each iteration $C$;
The total number of algorithm iterations $N$;
The frequency of training the online RL agent $Freq$;
Initial branching policy $\pi_0$;
Collect all samples generated in each iteration $D$
Collect the samples generated in the $i$-th iteration $D_i$
**Output**: Branch policy based on Hybrid RL $\pi_N$;

1: Let $i = 1$;
2: Let $D = \emptyset$;
3: **while** $i \leq N$ **do**
4:     Let $D_i = \emptyset$
5:     Let $t = 1$;
6:     **if** $i == 1$ or $i \% Freq == 0$ **then**
7:         $f_{\theta_{on}} = ActorCritic(\pi_{i-1}, \text{expert})$
8:     **end if**
9:     **while** $t \leq C$ **do**
10:         $O_t, A_t, R_t, S_t^s, S_t^p = f_{\theta_{on}}(\pi_{i-1}, \text{expert})$;
11:         $D_i = D_i \cup \{O_t, A_t, R_t, S_t^s, S_t^p\}$;
12:         $t = t + 1$;
13:     **end while**
14:     **if** $i == 1$ **then**
15:         $f_{\theta_{off}} = BDRL(D_i)$;
16:     **end if**
17:     Remove samples from $D_i$ as Eq. 4;
18:     $D = D \cup D_i$;
19:     $\pi_i = GCNN(D)$;
20:     $i = i + 1$;
21: **end while**

---

$$Tc^\top x^* - \int_0^\top d(t)^* dt \quad (6)$$

### 4.2 COMPARATIVE EXPERIMENT

**Baselines.** We compare our method with four baselines: **1) SCIP (v7.0.3)** (Gamrath et al., 2020a): State-of-the-art open-source solver with hybrid expert branching rules, named reliable pseudo cost branching. **2) GUROBI (v9.5.0)**: State-of-the-art commercial solver with hybrid expert branching rules which is known often much more effective than open-source solver. **3) GCNN** (Gasse et al., 2019): An IL-based model with graph convolutional networks for scoring. **4) ML4CO-KIDA** (Cao et al., 2022a): a Dagger-like (Ross et al., 1984) method based on IL, which is the best-performing method in the ML4CO 2021 competition.

**Training.** For iterative-based methods, including our proposed hybrid framework and ML4CO-KIDA, we generate 10,000 and 4,000 samples for training and validation at each iteration, respectively. As for the IL-based GCNN, we generate 100,000 samples for training and 40,000 samples for validation. Note that on the ML4CO datasets, we will use the model published by ML4CO-KIDA's author on balanced item placement and anonymous problems, and reproduce ML4CO-KIDA using the source code for the Workload Apportionment problem. The accuracy on testing instances for different problems is listed in Table 4, where (acc@n) denotes the accuracy of the selected branching variable by branching policy rules for sample generation, ranking top $n$ among the predictions.

**Comparative analysis.** Table 5 depicts the overall performance on four binary integer programming problems. For the easy instances, learning-based methods significantly outperform the expert rule-based policies, showing the effectiveness of the IL framework. Our method shows consistently higher

Table 3: The type and protocol of the binary integer programming problem.

| Difficulty | Combinatorial Auction | | Set Covering | | Independent Set | | Capacitated Facitlity | |
|---|---|---|---|---|---|---|---|---|
| | columns | rows | columns | rows | columns | rows | columns | rows |
| Easy | 500 | 195 | 1000 | 500 | 500 | 1948 | 10100 | 10201 |
| Medium | 1000 | 383 | 1000 | 1000 | 1000 | 3957 | 20100 | 20301 |
| Hard | 1500 | 591 | 1000 | 2000 | 1500 | 5938 | 40100 | 40501 |

Table 4: Evaluation of three IL-based methods by accuracy@$k$ on the test sets of seven problems.

| IL-based Methods | Combinatorial Auction | | Set Covering | | Independent Set | | Capacitated Facitlity | | Item Placement | | Workload Apportionment | | Anonymous | |
|---|---|---|---|---|---|---|---|---|---|---|---|---|---|---|
| | @1 | @10 | @1 | @10 | @1 | @10 | @1 | @10 | @1 | @10 | @1 | @10 | @1 | @10 |
| GCNN | 0.62 | 0.97 | 0.56 | 0.97 | 0.48 | 0.86 | 0.56 | 0.99 | 0.77 | 0.99 | 0.28 | 0.84 | 0.50 | 0.87 |
| KIDA | 0.58 | 0.96 | 0.55 | 0.97 | 0.41 | 0.86 | 0.56 | 0.99 | — | — | 0.23 | 0.78 | — | — |
| **HRL-Aug** | 0.55 | 0.96 | 0.52 | 0.97 | 0.40 | 0.85 | 0.56 | 0.99 | 0.73 | 0.99 | 0.42 | 0.88 | 0.48 | 0.89 |

Table 5: Instances are evaluated based on solving time, the number of wins (fastest method) versus the number of solved instances, and the number of nodes generated by B&B (lower is better).

| Methods | Easy | | | Medium | | | Hard | | |
|---|---|---|---|---|---|---|---|---|---|
| | Time | Win/Solved | Nodes# | Time | Win/Solved | Nodes# | Time | Win/Solved | Nodes# |
| SCIP | 2.42 ± 11.25% | 10/200 | **21.58** ± 39.64% | 32.92 ± 12.83% | 1/200 | 1446.18 ± 26.40% | 298.69 ± 8.70% | 5/200 | 30619.07 ± 9.78% |
| GCNN | 1.75 ± 9.73% | 47/200 | 78.54 ± 9.11% | 21.59 ± 9.87% | 63/200 | **1049.09** ± 5.35% | 225.23 ± 5.83% | 79/200 | 25864.09 ± 4.61% |
| KIDA | 1.70 ± 9.60% | 65/200 | 77.98 ± 8.92% | 21.62 ± 10.49% | 65/200 | 1064.76 ± 5.96% | 237.10 ± 6.25% | 16/200 | 27264.35 ± 4.87% |
| **HRL-Aug** | **1.68** ± 9.44% | **78**/200 | 77.40 ± 9.23% | **21.27** ± 10.75% | **71**/200 | 1053.42 ± 7.28% | **222.758** ± 6.81% | **100**/200 | **24218.31** ± 5.41% |

Combinatorial Auction

| Methods | Easy | | | Medium | | | Hard | | |
|---|---|---|---|---|---|---|---|---|---|
| | Time | Win/Solved | Nodes# | Time | Win/Solved | Nodes# | Time | Win/Solved | Nodes# |
| SCIP | 12.38 ± 9.94% | 3/200 | 215.20 ± 30.29% | 36.83 ± 5.68% | 0/200 | 902.26 ± 18.08% | 552.67 ± 3.76% | 52/170 | 50192.55 ± 5.53% |
| GCNN | **7.85** ± 8.54% | 75/200 | **190.47** ± 14.66% | 24.95 ± 7.51% | 70/200 | 692.89 ± 10.61% | 529.44 ± 3.74% | 30/170 | 34442.69 ± 3.92% |
| KIDA | 7.89 ± 8.04% | **79**/200 | 193.47 ± 14.71% | 25.47 ± 10.74% | 55/200 | 706.31 ± 15.11% | 526.01 ± 4.33% | 34/170 | **33880.34** ± 7.06% |
| **HRL-Aug** | 8.15 ± 8.85% | 43/200 | 201.76 ± 15.62% | **24.64** ± 8.36% | **75**/100 | **675.76** ± 11.67% | **525.42** ± 3.06% | **54**/170 | 34858.74 ± 7.56% |

Set Covering

| Methods | Easy | | | Medium | | | Hard | | |
|---|---|---|---|---|---|---|---|---|---|
| | Time | Win/Solved | Nodes# | Time | Win/Solved | Nodes# | Time | Win/Solved | Nodes# |
| SCIP | 9.86 ± 16.06% | 19/200 | **72.21** ± 47.65% | 121.17 ± 21.41% | 1/200 | 1168.90 ± 24.32% | 2048.93 ± 16.97% | 0/122 | 11969.63 ± 28.10% |
| GCNN | 7.35 ± 14.38% | 60/200 | 80.23 ± 32.02% | 56.85 ± 15.28% | 39/200 | 532.34 ± 20.90% | 767.95 ± 16.71% | 10/182 | 8519.85 ± 28.92% |
| KIDA | 7.39 ± 14.78% | 57/100 | 84.20 ± 31.01% | 52.88 ± 14.50% | **82**/200 | **483.53** ± 18.97% | 677.15 ± 15.01% | 48/185 | 8055.94 ± 19.74% |
| **HRL-Aug** | **7.34** ± 13.22% | **64**/100 | 82.00 ± 29.41% | **52.75** ± 12.50% | 78/200 | 508.79 ± 19.86% | **577.79** ± 14.77% | **137**/195 | **6048.33** ± 22.39% |

Maximum Independent Set

| Methods | Easy | | | Medium | | | Hard | | |
|---|---|---|---|---|---|---|---|---|
| | Time | Win/Solved | Nodes# | Time | Win/Solved | Nodes# | Time | Win/Solved | Nodes# |
| SCIP | 96.79 ± 19.25% | 27/200 | **591.01** ± 31.53% | 403.40 ± 17.69% | 41/200 | **741.34** ± 22.55% | 921.68 ± 22.59% | 37/191 | **392.67** ± 22.59% |
| GCNN | 60.89 ± 33.29% | 60/200 | 1100.35 ± 37.27% | 345.56 ± 29.52% | 47/200 | 1374.09 ± 31.95% | 769.84 ± 23.17% | 45/184 | 758.56 ± 25.13% |
| KIDA | 60.64 ± 34.08% | 52/200 | 1036.61 ± 37.23% | 349.1 ± 34.70% | 45/200 | 1403.62 ± 34.22% | 728.94 ± 22.61% | 57/184 | 736.21 ± 24.42% |
| **HRL-Aug** | **60.19** ± 30.02% | **61**/100 | 1027.54 ± 34.19% | **324.10** ± 28.49% | **67**/200 | 1348.08 ± 26.53% | **718.62** ± 20.54% | **58**/186 | 717.00 ± 19.31% |

Capacitated facility location

Table 6: Eperiment on the dataset of the ML4CO 2021 competition.

| Methods | item placement | | | Workload Apportionment | | | anonymous | | |
|---|---|---|---|---|---|---|---|---|---|
| | Dual Bound | Dual Gap | Reward | Dual Bound | Dual Gap | Reward | Dual Bound | Dual Gap | Reward |
| SCIP | 5.25 | 63.24% | —— | 699.36 | 3.17% | —— | 29926.57 | 52.10% | —- |
| GCNN | 5.18 | 61.19% | 4387 | **701.82** | 2.84% | 631127 | 31011.52 | 46.75% | 27344112 |
| KIDA | 8.719 | 34.18% | 7532 | 701.79 | 2.84% | 631147 | 31102.45 | 46.16% | 27548468 |
| **HRL-Aug** | **8.725** | **34.16**% | **7538** | 701.80 | **2.84**% | **631203** | **31170.81** | **45.64**% | **27672243** |

solving efficiency with quite close performance for easy problems. Interestingly, the results reveal that there is no direct connection between the final node counts and the solving time.

Table 6 gathers the results for ML4CO datasets. ML4CO-KIDA significantly outperforms GCNN, which reveals that iterative-based methods may deal with some challenges faced by pure IL models. Among all the competing baselines, our hybrid framework consistently dominates the others, illustrating the effectiveness of online RL which can drive much deeper exploration from the iteratively updated policies. Notably, the ablation study shows our proposed method can even outperform Gurobi on item placement problems, purely assisted by the open-source solver SCIP.

**Generalization analysis.** We further test our method on some difficult instances (Medium and Hard) to evaluate the generalization ability, and the overall results are gathered in Table 5. As can be seen, the default SCIP branching strategies significantly underperform the learning-based policies on solving time, revealing the limitations of the expert rules, especially on large-scale instances. Similarly, our proposed method consistently outperforms the competing baselines in terms of solving efficiency, showing a better generalization ability to larger instances.

## 4.3 ABLATION STUDY

We present an ablation study to verify the effectiveness of the online RL agent and offline RL agent. Specifically, we evaluate our method and ML4CO-KIDA, with different iteration steps.

In Table 7 and Table 8, the training time is categorized into sample generation time and model training time. For the training sample generation time, our approach closely aligns with that of ML4CO-KIDA

Table 7: Ablation study on Combinatorial Auction.

| iteration | HRL-Aug | | | ML4CO-KIDA | | |
|---|---|---|---|---|---|---|
| | Samples | Samples generation time | Training time | Samples | Samples generation time | Training time |
| 1 | 7,067 | 29 min | 19 min | 10K | 35 min | 11 min |
| 10 | 70,620 | 65 min | 59 min | 100K | 71 min | 78 min |
| 20 | 127,757 | 98 min | 129 min | 200K | 111 min | 175 min |
| 30 | 194,683 | 143 min | 201 min | 300K | 149 min | 266 min |
| 40 | 268,013 | 189 min | 257 min | 400K | 191 min | 359 min |

Table 8: Ablation study on anonymous.

| iteration | HRL-Aug | | | ML4CO-KIDA | | |
|---|---|---|---|---|---|---|
| | Samples | Samples generation time | Training time | Samples | Samples generation time | Training time |
| 1 | 6,627 | 1.78 hours | 1.36 hours | 10K | 1.68 hours | 0.26 hours |
| 10 | 68,235 | 19.30 hours | 9.57 hours | 100K | 16.67 hours | 6.27 hours |
| 20 | 136,756 | 38.92 hours | 18.28 hours | 200K | 40.35 hours | 29.45 hours |
| 30 | 20,3754 | 58.31 hours | 27.32 hours | 300K | 66.96 hours | 55.10 hours |
| 40 | 27,1761 | 77.09 hours | 36.33 hours | 400K | 83.85 hours | 59.20 hours |

Table 10: Further validating the generalizability of **HRL-Aug** in Combinatorial Auction.

| Methods | MLP | | | CNN | | | RNN | | |
|---|---|---|---|---|---|---|---|---|---|
| | Time | Win/Solved | Nodes# | Time | Win/Solved | Nodes# | Time | Win/Solved | Nodes# |
| SCIP | $2.36 \pm 10.41\%$ | 11/200 | $\mathbf{21.58} \pm 39.64\%$ | $2.34 \pm 10.56\%$ | 11/200 | $\mathbf{21.58} \pm 39.64\%$ | $2.32 \pm 10.65\%$ | 25/200 | $\mathbf{21.58} \pm 39.64\%$ |
| KIDA | $1.50 \pm 7.71\%$ | 83/200 | $98.63 \pm 8.68\%$ | $1.54 \pm 6.89\%$ | 86/200 | $102.94 \pm 9.02\%$ | $1.89 \pm 9.25\%$ | 73/200 | $94.10 \pm 8.73\%$ |
| Ours | $\mathbf{1.49} \pm 6.81\%$ | **106**/200 | $100.69 \pm 9.25\%$ | $\mathbf{1.53} \pm 6.90\%$ | **103**/200 | $100.18 \pm 8.42\%$ | $\mathbf{1.86} \pm 8.70\%$ | **102**/200 | $94.19 \pm 9.16\%$ |

on both the combinatorial auction and anonymous problem. However, with our newly incorporated offline RL agent, the training sample size of our proposed **HRL-Aug** reduced by around 30% at each iteration compared with ML4CO-KIDA. In general, the overall model training time was reduced by 28%, and 38% on combinatorial auction and anonymous problems, respectively, thus illustrating the efficacy and essential role of the offline RL agent.

Table 9 ablates the effect of online/offline RL for item placement. **HRL-Aug**-on means disabling the offline agent, and thus the training samples are not filtered by their qualities. **HRL-Aug**-off means disabling the online agent, and thus the expert experience is only considered. The results suggest that the online part is more important, and both parts have positive effects.

Table 9: Results on item placement.

| Methods | Dual Bound | Dual Gap | Reward |
|---|---|---|---|
| GCNN | 5.183 | 61.19% | 4387 |
| GUROBI | 8.522 | 36.65% | —— |
| **HRL-Aug**-off | 7.245 | 43.49% | 5983 |
| **HRL-Aug**-on | 8.417 | 37.35% | 7241 |
| **HRL-Aug** | **8.725** | **34.16**% | **7538** |

### 4.4 GENERALIZABILITY OVER VARIABLE SELECTION POLICY EMBODIMENT

To assess our methodology's network-wise generalization, we position it within the framework of a combinatorial auction problem. The validation process was rigorously executed on a set of easy-level problems, comprising 40 instances. Each instance underwent testing under five seeds. To verify the adaptability and generalization potential of our approach across diverse network architectures, we replaced GCNN with alternative models such as the Multilayer Perceptron (MLP, input features=17, hidden size=64, output features=1), Convolutional Neural Networks (CNN, input channels=17, output channels=64, kernel size=3), and Recurrent Neural Network (RNN, input size=17, hidden size=64, num layers=1, sequence length=500) as variable selection policies for both our method and ML4CO-KIDA. A comprehensive depiction of performance metrics has been thoughtfully compiled and presented in Table 10. The outcomes are unequivocal – our approach consistently outperforms both ML4CO-KIDA and SCIP. This observation substantiates the inherent model-agnostic nature of our methodology, emphasizing its versatility and effectiveness across various scenarios.

## 5 CONCLUSION AND OUTLOOK

We have proposed a hybrid online and offline RL approach to enhance the branching policy via efficient training set augmentation. Hybrid agents perform in an iterative manner, with an online RL agent deciding sample generation, either from the expert rule or learning-based policy, and an offline RL agent further filtering the generated samples with high cumulative returns. Experiments on different MIP problems show its effectiveness, even achieving superior performance over the leading commercial solver in some cases. **HRL-Aug** also shows superior generalization ability. Future work may combine learning to presolve (Liu et al., 2024) and instance generation (Wang et al., 2023; Li et al., 2023) with learning-based solvers in a synergetic manner.

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

# A  APPENDIX

## A.1  THE FEATURE OF GCNN

The bipartite graph features used throughout this paper are listed in Table 11.

Table 11: Description of the constraint, edge, and variable features in the GCNN

| Name | Feature | Description |
|---|---|---|
| Constraint features | obj_cos_sim | Cosine similarity with the objective. |
| | bias | Bias value, normalized with constraint coefficients. |
| | is_tight | Tightness indicator in LP solution. |
| | dualsol_val | Dual solution value, normalized. |
| | age | LP age, normalized with total number of LPs. |
| Edge features | coef | Constraint coefficient, normalized per constraint. |
| Variable features | type | Type (binary, integer, impl. integer, continuous) as a one-hot encoding. |
| | coef | Objective coefficient, normalized. |
| | has_lb | Lower bound indicator. |
| | has_ub | Upper bound indicator. |
| | sol_is_at_lb | Solution value equals lower bound. |
| | sol_is_at_ub | Solution value equals upper bound. |
| | sol_frac | Solution value fractionality. |
| | basis_status | Simplex basis status (lower, basic, upper, zero) as a one-hot encoding. |
| | reduced_cost | Reduced cost, normalized. |
| | age | LP age, normalized. |
| | sol_val | Solution value. |
| | inc_val | Value in incumbent. |
| | avg_inc_val | Average value in incumbents. |

## A.2  BRANCHING

Branching is the core of the branch-and-bound algorithm, and designing effective strategies was critical to mixed Integer programming (MIP) solving right from the beginning.

The overall pipeline for branching is listed as follows:

---

**Algorithm 2** Generic branching (variable selection)

---

1: **Input:** Current subproblem $Q$ with an optimal linear programming (LP) solution $\check{x} \notin X_{MIP}$
2: **Output:** An index $j \in I$ of an integer variable $x_j$ with the fractional LP value $\check{x} \notin \mathbb{Z}$
3: Let $F = \{j \in I | \check{x} \notin \mathbb{Z}\}$ be the set of branching candidates.
4: For all candidates $j \in F$, calculate a score value $s_j \in \mathbb{R}$
5: **return** an index $j \in F$ with $s_j = max_{k \in F}(s_k)$

---

There are many rule-based expert strategies, such as strong branching (Applegate et al., 1995), active constraint (Patel & Chinneck, 2007), and so on. For example, the widely utilized strong branching will select the fractional variable with the best dual bound improvement to branch on, listed as Eq. 7.

$$s_j = max\{\check{c}_{Q_j^-} - \check{c}_Q, \check{c}_{Q_j^+} - \check{c}_Q\} \tag{7}$$

where $s_j$ denotes the score for each fractional candidate $j$, $Q_j^-$ represents the left subproblem by adding the trivial inequality $x_j \leq \lfloor \check{x}_j \rfloor$, and $Q_j^+$ represents the right subproblem by adding the trivial inequality $x_j \geq \lceil \check{x}_j \rceil$. $\check{c}$ denotes the objective value of the LP relaxations.

For the GCNN policy, at each variable selection iteration, it will get the input features listed in Table 11, and predict $s_j$ for each fractional candidate variable. Then it will select an index $j \in F$ with $s_j = max_{k \in F}(s_k)$ as the branching variable.

### A.3 Difference in methodology and performance with ML4CO-KIDA (Cao et al., 2022a)

KIDA is a Dagger-like method based on IL. Though interacting with the solver with 95% probability of using model $\pi_{i-1}$ and 5% probability of using Strong Branching in the solving process, it only collected training samples from Strong Branching. While in our paper, there are the following major differences, which are also our contributions:

**1)** In the solving process, the variable selection from $\pi_{i-1}$ or strong branching was not decided by pre-defined parameters (95% vs 5% in KIDA). We proposed an online RL agent to dynamically determine the selection and may generate more reasonable samples. You can see Section 3.2 for details.

**2)** Different from KIDA, samples from strong branching and $\pi_{i-1}$ will both be collected by the decision from the online agent.

**3)** We devised an extra offline learning phase to filter higher cumulative reward samples, which can significantly reduce the training complexity and may improve imitation learning performance to some extent.

Besides, the data augmentation mechanism in the proposed methodology can be segmented into collectors and filters. The KIDA has no filtering phase. Key distinctions exist in the collection phase:

**1) Selection Mechanism:** In the case of KIDA, its selector operates under the influence of a Bernoulli distribution, wherein a hyperparameter is determined based on an in-depth grasp of domain expertise. Nonetheless, this approach grapples with manageability issues, particularly when the decision problem grows in complexity. Our approach takes a different path, circumventing this challenge. We leverage an online reinforcement learning agent that imbibes knowledge from data iteratively, thereby gradually converging toward an optimal distribution. This distribution is notably more intricate than the Bernoulli model, especially tailored to effectively tackle intricate decision problems.

**2) Collection Mechanism:** In the realm of KIDA, its collector solely acquires samples from the expert policy of strong branching. This policy, albeit optimal at the immediate juncture, might not necessarily hold the same effectiveness over an extended trajectory. Our approach, on the other hand, goes beyond this confined paradigm. We amass samples not just from the expert policy, but also from the model policy. This judicious balance between exploitation and exploration forms the bedrock of our imitation learning, propelling us to converge toward the zenith of the optimal distribution. This distribution, then, serves as the compass guiding our branching decisions, distinctly setting us apart from KIDA.

### A.4 Further Discussion on the Novelty of Our Data Augmentation Framework for Imitation Learning

Our primary innovation lies in the introduction of the data augmentation Framework, which is meticulously designed for the realm of imitation learning. This framework comprises three pivotal components: an online Reinforcement Learning (RL) agent, an offline RL agent, and an imitation learning-based policy. Refer to Figure 1 for a comprehensive depiction of this framework's architecture and functionality.

**1) The Online RL Agent:** The online RL agent plays a pivotal role by facilitating the profound exploration of uncharted scenarios. By striking a balance between exploration and exploitation, it effectively steers clear of local optimal solutions for branching variables. Consequently, the presence of the online agent substantially enhances the overall effectiveness of imitation learning.

**2) The Offline RL Agent:** As evidenced by Table 7 and Table 8, the primary advantage of the offline RL agent lies in its ability to enhance the efficiency of imitation learning. Remarkably, it accomplishes this by reducing the model training time by approximately 30%, so the model training time was reduced by 28%, and 38% in combinatorial auction and anonymous problem respectively. Thereby illustrating the efficacy and efficiency of the offline RL agent.

**3) The Imitation learning-based policy:** A crucial validation of our proposed Data Augmentation Framework (**HRL-Aug**) unfolds within the context of branching scenarios. At the core of this validation lies the deployment of an imitation learning-based policy. This policy, meticulously learned

Table 12: Compare with the RL for branching method.

| Methods | Easy | | | Medium | | |
| --- | --- | --- | --- | --- | --- | --- |
| | Time | Win/Solved | Nodes# | Time | Win/Solved | Nodes# |
| SCIP | $8.14 \pm 6.99\%$ | 4/200 | **121.78** $\pm 28.19\%$ | $29.39 \pm 5.53\%$ | 1/200 | $660.80 \pm 12.91\%$ |
| Scavuzzo et al. (2022) | $8.88 \pm 15.96\%$ | 8/200 | $1215.20 \pm 26.90\%$ | $63.05 \pm 18.24\%$ | 0/200 | $6192.78 \pm 21.82\%$ |
| Parsonson et al. (2023) | $8.86 \pm 9.60\%$ | 2/200 | $496.3 \pm 16.92\%$ | $50.67 \pm 12.08\%$ | 0/200 | $2585.4 \pm 14.68\%$ |
| **HRL-Aug** | **5.40** $\pm 8.84\%$ | **186**/200 | $149.34 \pm 7.37\%$ | **19.54** $\pm 6.77\%$ | **199**/100 | **555.61** $\pm 6.52\%$ |

Set Covering

through the imitation learning process, emerges as a pivotal component capable of supplanting expert-based policies within solvers, akin to SCIP as exemplified in this study.

Our main contribution is the novel data augmentation framework for imitation learning. We have indeed undertaken comprehensive tests on the generalization capabilities of our proposed framework. Specifically, we have replaced the learning-based Graph Convolutional Network (GCN) with alternative architectures, such as Multi-Layer Perceptrons (MLPs), Convolutional Neural Networks (CNNs), and Recurrent Neural Networks (RNNs) as shown in section 4.4. Our approach consistently outperforms both ML4CO-KIDA and SCIP. This observation substantiates the inherent model-agnostic nature of our methodology, emphasizing its versatility and effectiveness across various scenarios.

## A.5 COMPARE WITH THE RL FOR BRANCHING METHOD

Recently RL-based methods have been proposed to enhance the branching policies, which do not rely on the expert rule, and might find a better policy for branching. To further validate the effectiveness of our method, we conducted an experiment in the set covering to compare with two RL-based methods, including Scavuzzo et al. (2022) and Parsonson et al. (2023). We use similar metrics as those in Gasse et al. (2019) and set SCIP's parameter the same with Scavuzzo et al. (2022), Note that we use the model published by Parsonson et al. (2023) and Scavuzzo et al. (2022), their model is trained in the 500 rows, 1000 columns, and 400 rows, 750 columns set covering problem respectively. Then, evaluation is performed for Easy(500 rows, 1000 columns) and Medium (1000 rows, 1000 columns) on 40 generated instances using five different seeds, which amounts to a total of 200 solving attempts per method. As shown in Table 12, the results show that our method is obviously better than the RL-based method both in solving time and nodes.

## A.6 COMPARE THE NUMBER OF NODES SOLVED TO OPTIMAL BETWEEN GCNN (GASSE ET AL., 2019) AND STRONG BRANCHING.

Strong Branching is the most efficient branching strategy in terms of the number of nodes in the B&B tree. To verify whether the learned GCNN (Gasse et al., 2019) can achieve a smaller B&B tree than strong branching or not, we perform experiments on the combinatorial auction problem with 200 easy difficulty instances, as shown in Table 13.

Specifically, we compare the number of nodes each branching policy solved to optimal–Wins (Nodes). The Win (Nodes) shows that the GCNN (Gasse et al., 2019) can perform better than Strong Branching in some instances. It empirically verifies that the quality of variable selection in strong branching can be further improved.

Table 13: Compare the node counts when solved to optimal.

| Methods | Wins(Nodes) |
| --- | --- |
| GCNN | 40/200 |
| Strong Branching | 174/200 |

