# OpenReview forum: "Towards Imitation Learning to Branch for MIP: A Hybrid Reinforcement Learning based Sample Augmentation Approach"
_ICLR.cc/2024/Conference — ICLR 2024 poster_

### Official Review · Reviewer_pX8W · 2023-10-16

**Soundness:** 3 good
**Presentation:** 4 excellent
**Contribution:** 2 fair
**Rating:** 6
**Confidence:** 5

**Summary:**

Update on November 20:

I raised my score to 6 based on the authors' responses. I am willing to keep discussing with the authors and the other reviewers to achieve a fully discussed final score.


---
This paper proposes a novel iterative collection and filtering framework to leverage a combination of online and offline RL techniques. Experiments show it consistently outperforms the previous SOTA approaches. Moreover, the authors claim that the proposed approach can be regard as a plugin orthogonal to peer methods.

**Strengths:**

1. The experiment is thorough. The authors provide very detailed empirical results to demonstrate the effect of the proposed approach.
2. Clear writing, the paper is clearly structured and easy to go through flow.
3. The proposed approach is technically sound. The employment of RL in this task is technically sound.

**Weaknesses:**

1. Concerns about the motivation. Personally, I am not convinced about the motivation to introducing RL in the branching task. As reported in some previous work [1,2], the number of expanding nodes of FSB is significantly less than that of SOTA ML approaches, indicating that *FSB is a good enough expert policy to imitate*. Instead, the bottleneck in this topic may mainly lie in the low IL accuracy. The lower accuracy may due to the unsatisfactory model structure or the insufficient information in the widely-used bipartite graph states (observations, more precisely). However, currently I found no definitive answer about that in recent researches. The ablation study in a recent research [3] gives clues that the historical information is effective as the process is a POMDP. Thus, in my personal opinion, the latter reason seems closer to the truth. Thus, based on these results, I think all researches that proposing more complex online RL framework is somewhat incremental, as improving the IL accuracy seems to be the key.
2. Concerns about the unnecessary complexity for introducing RL. Though usually higher asymptotic performance, many RL approaches are usually sensitive to the hyperparameters, making their application requiring much manual tunings. I am doing research in both RL and CO, empirically, I found their combinations can be fragile. In this paper, the online RL is mainly used for data collection. However, as far as I know, RL approaches are usually sensitive to the data distribution due to the deadly triad. For example, TD3 [4] may fail in MuJoCo tasks when the initial 20k random steps of data collection is turned off; BCQ [5] may fail when the offline data are collected with hybrid policies. Thus, in real-world CO tasks, I prefer use simple GNN+IL approach rather than other complex but fragile approaches, even if they claim higher performance in the four synthetic benchmarks. For this paper, I am concerned about the complexity for introducing RL.
3. More explanations is required. In this paper, the RL based approach achieves lower IL accuracy while higher e2e performance. Thus, I am interested in what kind of multi-step information it learns from the input states. I believe this is more meaningful than simply reporting performance improvements, as it can guide us in designing better input features as mentioned in Point 1. If only the approach is given but the explanations is missed, then this paper is more like just a application of existing RL approaches to a new "Atari".
4. Limited improvement compared to ML4CO-KIDA. The performance of HRL-Aug reported in Table 7 seems to be marginal to ML4CO-KIDA.

Anyway, the comments above may be too tough for studies in this research field. Thus, if the other reviewers all give more  positive scores, I will be willing to keep discussing with the authors and the other reviewers to achieve a fully discussed score.

[1] Gasse, Maxime, et al. "Exact combinatorial optimization with graph convolutional neural networks." Advances in neural information processing systems 32 (2019).

[2] Gupta, Prateek, et al. "Hybrid models for learning to branch." Advances in neural information processing systems 33 (2020): 18087-18097.

[3] Seyfi, Mehdi, et al. "Exact Combinatorial Optimization with Temporo-Attentional Graph Neural Networks." Joint European Conference on Machine Learning and Knowledge Discovery in Databases. Cham: Springer Nature Switzerland, 2023.

[4] Fujimoto, Scott, Herke Hoof, and David Meger. "Addressing function approximation error in actor-critic methods." International conference on machine learning. PMLR, 2018.

[5] Fujimoto, Scott, David Meger, and Doina Precup. "Off-policy deep reinforcement learning without exploration." International conference on machine learning. PMLR, 2019.

**Questions:**

1. Is the online RL training necessary? Intuitively, using an offline-trained policy to collect data and then update the policy with the collected data iteratively seems to be enough. Why is the online updating of the policy during data collection necessary?
2. What is the dual bound change when a child node is infeasible? In SCIP, the solver add the objective and the current global primal bound to the constraints, making the infeasible child appears in high frequency. Thus, in the calculation of FSB scores, the score of infeasible child is set to a dynamic large value. How about that in this paper?

---

> ### Author Response · Authors · 2023-11-15
>
> >***Q1:*** Concerns about the motivation. I think all research that proposes a more complex online RL framework is somewhat incremental, as improving the IL accuracy seems to be the key.
>
> Although Strong Branching is often used for imitation learning and the number of expanding nodes of FSB is significantly less than that of SOTA ML approaches. But it is a rule-based method, which means it tests which of the fractional candidates gives the best progress in the dual bound before actually branching on any of them, which can be viewed as finding the locally best variable to branch on [1]. This means the quality of variable selection in Strong branching can be further improved. So, our method aims to further improve the quality of training samples for imitation learning and jump out from the locally best variable selection strategy of expert rules. As you mentioned, improving the IL accuracy means the exploitation is much higher, but exploration is also important for imitation learning, so how to balance exploitation and exploration also can improve the performance of imitation learning, because in the previous research [2] it has been proven that adding past good experiences can add exploration for imitation learning and may find a potentially better policy than only imitating the expert policy.
>
> >***Q2:*** Concerns about the unnecessary complexity of introducing RL and the online updating of the policy during data collection.
>
> To further enhance the efficiency of imitation learning, our approach focuses on the decision of whether to include past good experiences generated by the previously learned policy in the training samples. To achieve this, we employ an online RL agent in each variable selection to make this decision. However, as the past-learned policy iteratively improves, it is crucial to periodically update the online RL agent to ensure more accurate decision-making. Alongside the online RL agent, we also train an offline RL agent as part of our methodology. The offline RL agent serves two purposes: improving the quality of training samples and reducing training time for imitation learning. By leveraging the offline RL agent, we can filter and select higher-quality training samples, leading to improved learning performance.
>
> In terms of experimental results, our method consistently outperforms state-of-the-art learning-based models in synthetic benchmarks, as well as in three ML4CO datasets, as demonstrated in Table 6. Moreover, our approach significantly reduces the training time compared to ML4CO-KIDA, as shown in Table 7 and Table 8. Overall, our approach combines the use of an online RL agent for decision-making and an offline RL agent for sample filtering. Through these enhancements, our method achieves superior branching decisions. Additionally, our approach effectively reduces training time compared to the ML4CO-KIDA method.
>
> >***Q3:*** More explanations about the RL-based approach achieving lower IL accuracy while higher e2e performance is required.
>
> As motivation mentioned in Q1, our method focuses on balancing exploitation and exploration to improve imitation learning. The biggest difference between our method and GCNN, and ML4CO-KIDA methods is the training samples, which are combined with input features and labels. For the input feature, all the methods use the bipartite graph feature. For the label, our method's use of the online RL decides a better label in each variable selection, so, our method can generate high-quality training samples rather than only collecting samples from experts. Besides, the offline RL agent also filters higher cumulative reward samples generated from the online RL phase, which not only improves the quality of training samples but also reduces training time for imitation learning. with the iteration, the training sample and learned branching policy will be further optimized. Interestingly, Table 4, Table 5, and Table 6 show that the accuracy of the different methods, namely our method, GCNN, and ML4CO-KIDA, does not correlate with their overall performance.
>
> >***Q4:*** Limited improvement compared to ML4CO-KIDA.
>
> As shown in Table 5, as the difficulty of the problem increased, in the easy and medium difficulty, the improvement over ML4CO-KIDA was marginal. But in the hard difficulty, especially in the Maximum Independent Set problem, the improvement becomes more obvious. So, we further evaluate our method in the ML4CO datasets in Table 6, our method's reward dominates the ML4CO-KIDA in all three datasets, especially in anonymous problems, which are inspired by real-world applications of large-scale problems.
>
> >***Q5:*** What is the dual bound change when a child node is infeasible?
>
> In this paper, the dual bound change when a child node is infeasible is consistent with your description.
>
> **References**:
>
> [1] Tobias Achterberg et al. Branching rules revisited. Operations Research Letters, 2005.
>
> [2] Junhyuk Oh et al. Self-imitation learning. In ICML, 2018.

---

> > ### Comment · Reviewer_pX8W · 2023-11-18
> > **Response to the Authors**
> >
> > Thank you for the long response. I appreciate your detailed reply to each of my question, though I still can not agree with some of the points in your reply. I will list my comments as follow for friendly discussions.
> >
> > ## About the response to Q1:
> >
> > I fully disagree with the authors' point on "the quality of variable selection in Strong branching can be further improved". The one-step optimal policy FSB, though might not be the optimal policy for long-term decisions, is good enough in the four synthetic benchmarks widely used in previous research. One key clue for this claim is that it obtains a B&B tree significantly smaller than all the other (both classical and learning-based) branching policies. The issue at hand is that all AI models, when trained with FSB demonstrations, fall far away from the performance of FSB. Thus, it is more likely that the bottleneck lies either in the expressive power of existing AI models, or in the the limited information of the bipartite graph inputs. Then, before AI models achieve similar performance to FSB, any research that claims they achieve high performance via improving the quality of the expert, is very confusing to me.
> >
> > If you still insist that improving the quality of expert demonstration makes sense, I personally recommend you to clarify the motivation of your work by considering the landscape of the learning objective. A good expert data is not always a good supervise learning signal. This perspective is more meaningful, as the learning process of RL algorithms, compared to direct imitation learning, is exactly a step-by-step learning process. This process might reduce the learning difficulty, which is also the core motivation of curriculum learning and knowledge distillation.
> >
> > ## About the response to Q2:
> > Based on my experience on RL, I still highly worry about the stability of incorporating multiple RL agents together. However, I have not evaluate it by myself, and thus I am not convinced about my intuition. Moreover, from my experience, when evaluate with more than 100 instances, the stand deviation of the end-to-end solving time on this task can reduce to a small range. However, the stand deviation in Table 5 seems to be relatively large. What might be the potential reason?
> >
> > ## About the response to Q3:
> > I agree with your point "the accuracy of the different methods does not correlate with their overall performance". This phenomenon can be observed in many real-world datasets.
> >
> > ## About the response to Q4:
> > Sorry for my carelessness. The improvement seems to be more significant on hard problems. Can you provide more explanations and intuitions to help readers better understand the essential reasons for this improvement?

---

> > > ### Author Response · Authors · 2023-11-19
> > >
> > > >***Q1.1:*** Further discuss the motivation about the quality of variable selection in Strong branching can be further improved.
> > >
> > > We appreciate your valuable suggestions. To verify this point, we here again provide evidence both from literature and our first-hand empirical experimental results as follows.
> > >
> > > **Literature**:
> > >
> > > The SCIP’s author [1] has proved that it can be viewed as finding the locally (with respect to the given score function) best variable to branch on, as we already argued in our main paper’s introduction section.
> > >
> > > To further improve its performance, SCIP developers further designed the lookahead branching [2], which considered more dual bound information from deeper levels to deal with the potential local search limitations of strong branching, rather than evaluating only the one-step performance in strong branching.
> > >
> > > Banitalebi-Dehkordi et al [3] also show that the effect of imitating pseudo-cost branching expert rules is much better than Strong branching in the ML4CO dataset, and even the imitation accuracy of the former is much lower than that of the latter.
> > >
> > > The ML4CO-KIDA [4], which is the best-performing method in the ML4CO, also shows that Strong Branching cannot produce completely reliable labels due to the dual degeneracy. The high dual degeneracy in the LP solution may cause the product score of Strong Branching to combine the improvements of the two child nodes close to zero.
> > >
> > > **Empirical experimental results**:
> > >
> > > As we know, Strong Branching is the most efficient branching strategy in terms of the number of nodes in the B&B tree, so, how to further enhance the training samples for imitation learning is a very challenging task. We deep dive into the test results in the combinatorial auction problem with easy difficulty, as shown in the following table.
> > >
> > > Specifically, we compare the number of nodes each branching policy solved to optimal--Wins (Nodes). The Win (Nodes) shows that the learning-based method can perform better than Strong Branching in some instances. It empirically verifies that ‘the quality of variable selection in Strong branching can be further improved’ and learning the past good experience can further improve imitation learning, which shows in the result that our method further improves the Wins(Nodes) Compared with GCNN.
> > >
> > > We appreciate your valuable suggestions. We will clarify this point in the final version.
> > >
> > > | Method | Wins(Nodes)             |
> > > | ------ | ---------------- |
> > > | Strong Branching | 173/200 |
> > > | GCNN       | 31/200     |
> > > | HRL-Aug    | 38/200 |
> > >
> > >
> > >
> > > >***Q1.2:*** Clarify the motivation of your work by considering the landscape of the learning objective.
> > >
> > > We really appreciate your valuable suggestions. What we proposed in this paper is a data augmentation framework, that can select from the best of the two worlds, expert rules or the past learned policy. In general, it can generalize to different scenarios to further enhance the expert rules in some cases, not limited to the variable selection in this paper. Specifically for the branching scenario, as you mentioned, though strong branching was already good enough proved by previous research, it may still fail in some specific cases, as evidenced by the Table above. To obtain a generally better branching policy, we proposed to combine the past learned policies with the experts, as they may deliver fewer nodes in some instances. In this respect, we designed the online RL agent to select from the two sources, where the generated samples may enhance the pure strong branching by the cumulative rewards when it, unfortunately, failed in some cases. The experimental results in Table 5 and Table 6 can further demonstrate our insights, as our proposed framework can consistently outperform the GCNN and KIDA, which only take the expert strong branching rules as imitation samples, across all the problems.
> > >
> > > **References**:
> > >
> > > [1] Tobias Achterberg et al. Branching rules revisited. Operations Research Letters, 2005.
> > >
> > > [2] Gleixner et al. The SCIP Optimization Suite 6.0[J].  2018.
> > >
> > > [3] Banitalebi-Dehkordi A et al. ML4CO: Is GCNN All You Need? Graph Convolutional Neural Networks Produce Strong Baselines For Combinatorial Optimization Problems, If Tuned and Trained Properly, on Appropriate Data[J].
> > >
> > > [4] Cao Z et al.ML4CO-KIDA: Knowledge Inheritance in Dataset Aggregation[J]. 2022.

---

> > > > ### Comment · Reviewer_pX8W · 2023-11-19
> > > >
> > > > Thank you a lot for the response and effort in addressing all of my comments. I also gained some new insight on this task from your detailed response. Your response partially addressed some of my concerns. Thus, I raised my score to 6.
> > > >
> > > > Currently, I still have several questions&comments:
> > > > 1. The reference [3] provided by you is very confusing to me. Since vanilla PB is not a time-consuming expert, what is the motivation to conduct imitation learning based on it? I roughly searched the key word "pseudo-cost" but did not find the claim mentioned by you (sorry for not reading it carefully due to the insufficient time). If this claim is correct, could this reason be that "pseudo-cost is easier to learn" rather than "pseudo-cost is a better expert" on these benchmarks?
> > > > 2. I personally recommend you to provide more insight and motivations (and their corresponding empirical illustrations) in your paper. Currently, this paper seems to be just a direct combination of several RL techniques to the branching task. But from my personal taste, the motivations and insights provided in your rebuttal is more crucial to this community.
> > > > 3. The "high dual degeneracy in the LP solution" is a new but reasonable perspective to me. I also observed that in the cut selection tasks.
> > > > 4. Your might provide an incorrect reference [5].
> > > >
> > > > I am willing to keep discussing with the authors and the other reviewers to achieve a fully discussed final score.

---

> > > > > ### Author Response · Authors · 2023-11-20
> > > > >
> > > > > >***Q1:*** What is the motivation to conduct imitation learning based on pseudo-cost Branching in reference [3].
> > > > >
> > > > > By diving deep into this reference carefully, we find it is the same as Gassel et al [5], they all only imitate the Strong Branching with a certain probability.
> > > > >
> > > > > >***Q2:*** The motivations and insights provided in your rebuttal are more crucial to this community.
> > > > >
> > > > > We appreciate your valuable suggestions. we have highlighted the motivations and insights in the updated PDF file, as shown in the introduction section with blue text.
> > > > >
> > > > > >***Q4:*** You might provide an incorrect reference [5].
> > > > >
> > > > > Thank you for your response, we fixed this in the current response.
> > > > >
> > > > > **References**:
> > > > >
> > > > > [5] Gasse, Maxime, et al. "Exact Combinatorial Optimization with Graph Convolutional Neural Networks." (2019).

---

> > > > > > ### Comment · Reviewer_pX8W · 2023-11-21
> > > > > >
> > > > > > Thank you for the response. Most of my questions have been answered. I will keep my current rating (6) and encourage the authors to further improve the paper according to my suggestions above.

---

> > > ### Author Response · Authors · 2023-11-19
> > >
> > > >***Q2:*** The stand deviation in Table 5 seems to be relatively large.
> > >
> > > Thank you for your detailed comments. The reason for the potentially large standard deviation is the more randomized settings. Specifically, we not only follow the protocols with Gassel et al [5] on the permuted order of variables and constraints, the random seed for LP solver, e.g., for perturbations in the simplex. The settings are shown as follows.
> > >
> > > ```
> > > m.setBoolParam("randomization/permuteconss", True)
> > > m.setBoolParam('randomization/permutevars', True)
> > > m.setIntParam('randomization/permutationseed', seed)
> > > m.setIntParam('randomization/randomseedshift', seed)
> > >
> > > m.setIntParam("randomization/lpseed", seed)
> > > ```
> > >
> > > >***Q4:*** The improvement seems to be more significant on hard problems.
> > >
> > > Previous research [6] has proved that past good experiences can add exploration for imitation learning and may find a potentially better policy than only imitating the pure expert policy. With the increase of the problem difficulties, more and more branching decisions have to be performed within the larger branch and bound tree. In these cases, exploration may play a vital role in dealing with the local search limitations of pure strong branching policy. Therefore, when applied to large-scale problems, our proposed framework can perform significantly better over the competing baselines that purely take expert strong branching rules as imitation labels.
> > >
> > > **References**:
> > >
> > > [5] Fujimoto S et al.Off-Policy Deep Reinforcement Learning without Exploration[J].  2018.
> > >
> > > [6] Junhyuk Oh et al. Self-imitation learning. In ICML, 2018.

---

### Official Review · Reviewer_Gdrc · 2023-10-30

**Soundness:** 3 good
**Presentation:** 2 fair
**Contribution:** 3 good
**Rating:** 6
**Confidence:** 3

**Summary:**

This paper presents a hybrid RL approach for variable selection in mixed integer programming (MIP). The proposed approach uses online RL to select between a rule-based expert and imitation learning based policy. The data collected from this online agent is filtered by a mechanism referred to as the offline agent and sent to a GCNN-based imitation-learning agent. This process repeats iteratively for a number of iterations. Results are presented on a number of integer programming problems and compared to an open-source and commercial solver with expert branching rules and two imitation learning methods. The proposed method outperforms the baselines on most domains and an ablation shows the importance of the proposed sub components.

**Strengths:**

Overall the results in the paper look strong. I do not have much experience with the MIP problem and cannot comment on the baselines but from the presentation they seem relevant and appropriate. From an RL and algorithmic perspective there are some additional ablations that could be interesting, but the presented ablations seem reasonable to me.

**Weaknesses:**

To me, the primary weakness in the current draft is the presentation. The ordering of the Tables in the Experiments section are confusing - tables are introduced and described in a non-sequential order and there are typos and odd sentences in the text that make the descriptions hard to follow at times. For example Table 3 and 4 show the Ablation results of comparing HRL-Aug to ML4CO-KIDA while the main results are presented later in Table 7 and 8. Just re-ordering the Table sequences would make it much easier to read.

I have a number of questions and clarifications relating to the algorithm which I will described under `Questions` but could also count as weaknesses.

Apart from that there are a few typos and textual clarifications I will list below:

    1. Page 4 sentence 1: ‘Framework’ should be ‘framework’.
    2. Section 3.3: off-policy methods should be ‘offline methods’.
    3. Section 4.1: Dataset has `We’ appear in many places which should be `we’.
    4. Section 4.4 mentions: “We genuinely appreciate the collaborative effort of the reviewers and their invaluable role in shaping our work’. This is an awkward phrase that isn’t warranted before the review process? To me, it seems likely this is a vestige of a resubmission.

**Questions:**

My questions mostly relate to the RL part of the proposed approach.

1. What agent was used to train RL online? Was it a REINFORCE-like algorithm or something more complex? This may be important because the convergence guarantees of RL algorithms are often made in the discounted setting with \gamma < 1. It is also important to specify the precise algorithm used for reproducibility.
2. I’m confused as to the ‘Offline RL’ agent. To me, offline RL implies learning a policy i.e. an action selection mechanism - using fixed trajectories of data that were generated by some behavior policy (or policies). In the paper, what is described is more of a filtration mechanism where effectively a function is fit to returns generated in the first iteration and then subsequently used to threshold the trajectories to sample. Could the authors clarify this point?
3. There are a number of choices made in the paper whose impact is not clear. For example, the filtration mechanism parameters are fit on the first iteration and kept constant after. Equation 4 uses a \lambda parameter to regularize the fit.The online agent is trained every 5 iterations (Freq). Some unspecified choice of `z’ defines how the trajectories are sub-sampled. It is not clear to me how important any of these choices are. Ideally the paper would include ablations on these but I can understand the difficulty of presenting so much information with a page limit. If the authors could indicate why these choices were made (with possible empirical evidence in the Appendix), I think it would make the paper stronger.

---

> ### Author Response · Authors · 2023-11-15
>
> >***Q1:*** The ordering of the Tables in the Experiments section is confusing and there are typos and odd sentences in the text.
>
> To make the readers follow the descriptions effectively, we have addressed these issues in the updated PDF file. We assure you that we will provide clearer explanations in the final version of the paper.
>
> >***Q2:*** What agent was used to train RL online?
>
> Our methods use the advantage actor-critic to train the online RL agent, which combines the Actor-Critic method and the concept of advantage function. The Actor is responsible for selecting an action based on the current state, while the Critic evaluates the value of this action. The advantage function represents the advantage of an action relative to other actions and can be used to evaluate the value of an action. So, this algorithm can better estimate the value of actions and update parameters based on these values.
>
>
> >***Q3:*** This may be important because the convergence guarantees of RL algorithms are often made in the discounted setting with $\gamma$ < 1.
>
> RL agents have traditionally been tasked with maximizing the value function of a MDP, either in continuous settings, with fixed discount factor $\gamma$ < 1, or in episodic settings, with $\gamma$ = 1 [1]. Because the branching scenario is the episodic setting and the dual bound may remain unchanged for hundreds of steps shown in Figure 2, we set the gamma = 1. Specifically, the value of the gamma does not directly determine the convergence of the model, which is a discount factor used to balance the reward of current and future action in the learning algorithm. But in the branching scenario, with the discount factor $\gamma$ < 1, the reward may have little contribution to the cumulative reward after thousands of branches, which may cause an inaccurate evaluation of the action.
>
> >***Q4:*** Confused with the Offline RL agent. To me, offline RL implies learning a policy, why offline RL is described as a filtration mechanism in this paper.
>
> In the context of branching scenarios, learning a branching policy using RL-based methods is less effective compared to imitation learning [2], also presented in Appendix A.5 of our paper. To address this challenge and improve the performance of imitation learning while reducing training time, our method incorporates the use of offline RL agents. This approach aligns with recent research [3], which explores the training of offline RL agents to filter higher cumulative reward training samples.
>
> In Section 3.3 of our paper, we provide a detailed explanation of the implementation. During the training phase, we formulate a constrained optimization problem (Eq.3) that takes input data $\{(O_t, A_t, R_t)\}$ collected from the online RL phase and outputs a real number to fit the cumulative reward. Once the offline RL agent has been trained, it can provide an optimal solution for this constrained optimization problem based on the input state. Concretely, when presented with a new state $\{(O_t, A_t, R_t)\}$, the offline RL agent predicts an optimal cumulative reward $f_{{\theta {{\rm{off}}}}}(O_t, A_t)$. We then utilize this prediction to filter out training samples with actual cumulative rewards below a certain threshold, such as $R_t\geq\ zf{{\theta _{{\rm{off}}}}}(O_t, A_t)$. By employing this approach, we effectively filter out lower cumulative reward samples, focusing on those with higher cumulative rewards.
>
> The integration of offline RL agents in this manner provides a mechanism to enhance the performance of imitation learning, reduce training time, and filter training samples based on higher cumulative rewards.
>
>
> >***Q5:*** The hyperparameter settings in this paper should further explanation.
>
> We will present how to set the hyperparameter in this paper, for the offline RL agent we only train it in the first iteration, because this agent is mainly used to delete the samples with lower cumulative reward, we want to keep it consistent filtration mechanism for the fellow iterations, and the hyperparameter 'z' is set such that the top p% of the data points are selected, so we set it with manual tunings to keep the remaining around 70% of samples, although some research is more radical than ours[3], hyperparameter $\lambda$ is set with 0.02 aims to interpolate the cumulative reward and avoid overfitting. However, for the online RL agents, with the past-learned policy iterative optimized, to generate training samples more accurately, we must update the online RL agent periodically, so, we set the Freq=5, because we observed their relatively higher improvement for the learning branching policy after five iterations.
>
> **References**:
>
> [1] Pitis, S. (2019). Rethinking the Discount Factor in Reinforcement Learning: A Decision Theoretic Approach. AAAI.
>
> [2] Lara Scavuzzo et al. Learning to branch with tree mdps. NeurIPS, 2022.
>
> [3] Qingyu Qu et al. Yordle: An efficient imitation learning for branch and bound. arXiv preprint arXiv, 2022a.

---

> > ### Comment · Reviewer_Gdrc · 2023-11-20
> > **Response to authors**
> >
> > Thank you for replying to my comments. I am glad the structure of the paper has been changed to more easily follow the experiments. I also agree with the AdvantagectorCritic algorithm being a reasonable choice for the online agent.
> >
> > My point regarding offline RL was not to clarify what the method does - I understand that the implementation collects data and fits a number based on which trajectories are filtered. I think this is precisely why this is better referred to as a filtration mechanism rather than offline RL. As I stated before, offline RL implies learning a policy using the stored data buffer. Currently the approach trains an optimal cumulative reward function - can we assume this is equivalent then to a value function for f(O_t, A_t) under an optimal policy? My issue is purely one of naming: I think it isn't a big problem but potentially misleading to call filtering of trajectories offline RL here.
> >
> > Regarding the hyperparameter settings: my point was that there are no figures in the paper describing and rigorously evaluating the shoices made. I'm sure the higher performance occurs with a Frequency of 5 but an empirical validation of this is what I was hoping for.
> >
> > Thanks again for the comments and updating the paper.

---

> > > ### Author Response · Authors · 2023-11-21
> > >
> > > >***Q1:*** My issue is purely one of naming: I think it isn't a big problem but potentially misleading to call filtering of trajectories offline RL here.
> > >
> > > We appreciate your feedback regarding the potential confusion surrounding offline RL. To make it clear, we added a footnote on its first appearance in the updated PDF file: *Offline RL is a filtration mechanism similar to [1]*.
> > >
> > > >***Q2:*** Regarding the hyperparameter settings: My point was that there are no figures in the paper describing and rigorously evaluating the choices made.
> > >
> > > We appreciate your valuable suggestions, which may further improve our method. While the discussion time window is very tight and the experiments for the hyperparameters are very time-consuming, we will clarify the results in the final version.
> > >
> > > [1] Chen X , Zhou Z , Wang Z ,et al.BAIL: Best-Action Imitation Learning for Batch Deep Reinforcement Learning[J].  2019.DOI:10.48550/arXiv.1910.12179.

---

### Official Review · Reviewer_f3Ne · 2023-11-01

**Soundness:** 2 fair
**Presentation:** 2 fair
**Contribution:** 3 good
**Rating:** 5
**Confidence:** 4

**Summary:**

This paper studies the use of imitation learning (IL) and Reinforcement Learning (RL) to tackle Mixed Integer Programming (MIP) problems. The main contribution of the paper is a framework that is specifically designed for MIP problems. Empirical results on a number of MIP problems show that the proposed method can achieve good performance with reduced model training time.

**Strengths:**

**originality**
- The paper's main novelty is a new framework that is designed to use IL and RL to tackle MIP problems.

**quality**
- The overall presentation is good
- The paper discusses related works in a fairly clear manner

**clarity**
- Overall the paper is clear and easy to follow, however, the explanation on how the agent works can be improved

**significance**
- The paper studies how IL and RL can be applied to the MIP problems, which seems to be an important research direction
- The improved training time and better performance can be a significant result.

**Weaknesses:**

Discussion on the proposed method:
- I find the writing on how the proposed method works is a bit confusing, this might be partly due to the complexity of the problem. For example, what exactly actor critic method did you use in the online setting (for example, A type of SAC? DQN?)? And how exactly does the online RL agent decide whether to use the expert or to use the learned policy? (if for example, you say the action space is discrete, with 2 actions, one to choose the expert and the other to choose the learned policy for the online RL agent and it is a Q-learning type of agent, then it becomes much clearer) Currently I don't fully follow what is happening here. (also see questions)
- Another concern is the proposed method seems to only apply to only the MIP problems, and although the empirical results are interesting, it is a little unclear to me how much technical novelty is in the design of this framework and whether the contributions in this paper is significant enough.

**Questions:**

- How time consuming is strong branching compared to your method? Can you provide a wall-clock time comparison?
- It is a bit unclear to me how exactly does the online agent decide whether to use the learned policy or expert policy?
- Page 5, section 3.3, "batch DRL" is a general concept and is essentially the same as "offline DRL", which is initially discussed in some earlier papers (e.g. "Off-policy deep reinforcement learning without exploration" by Fujimoto et al.). The method in Chen et al., 2020 (Best Action Imitation Learning) is one of the imitation learning methods to tackle the batch/offline DRL problem. You might want to change the writing here to be more accurate.
- What is the "ActorCritic" algorithm you are using?
- Table 6 why there is no highlight on the best performing method for each task? Or the values are not related to performance?- Typically RL agents can take time to train, why is it the case that the proposed method, despite its complexity and a multi-stage/agent setup, can reduce training time compared to other methods?

---

> ### Author Response · Authors · 2023-11-15
>
> >***Q1:*** The explanation of how the online agent works and how exactly does the online agent decide whether to use the learned policy or expert policy?
>
> We really appreciate your valuable suggestion. Our methods use the advantage actor-critic (A2C) to train the online RL agent, which combines the Actor-Critic method and the concept of advantage function. The Actor is responsible for selecting an action (the action space is discrete, with 2 actions, one to choose the expert and the other to choose the learned policy) based on the current state. At the same time, the Critic evaluates the value of this action. The advantage function represents the advantage of an action relative to other actions. It can be used to evaluate the value of an action. Then use the cumulative reward to measure the quality of the agent's behavior in a given state. So, this algorithm can better estimate the value of actions and update parameters based on these values. When the online RL agent is trained, as shown in Figure 1, in each variable selection, input the state $s_t^O$ to the online RL agent, which will output an action 0 or 1, representing collect the sample generated from past-learned policy or expert policy. It is expected to make better decision-making in each variable selection, generate higher-quality training samples, and achieve effective exploration for imitation learning. This way, the sample generation process can be balanced between exploration and exploitation.
>
> >***Q2:*** It is a little unclear to me how much technical novelty is in the design of this framework.
>
> In Appendix A.4, we further discuss the Novelty of Our Data Augmentation Framework for Imitation Learning. Specifically, our framework not only proposed a new approach to enhancing the effectiveness of imitation learning, which is to better balance the exploration and exploitation of imitation learning but also improved the training efficiency of imitation learning, overall, this is a novel data augmentation framework and is technically nontrivial.
>
> We also have undertaken comprehensive tests of our proposed framework. Extensive experiments on different MIP problems show the effectiveness of our method compared with open-source solver SCIP and other competing baselines, even achieving superior performance over the leading commercial solver in some cases. In addition, our method shows superior generalization ability.
>
> >***Q3:*** How time-consuming is strong branching compared to your method?
>
> The strong branch is often used for imitation learning because it is a globally effective yet computationally expensive rule. At the beginning of our experiment, we also compared with the strong branch rule, as shown in the table below, we tested the result in the combinatorial auction problem with easy difficulty. However, as the experiment progressed, we found the strong branch was too slow, and it seriously affected the efficiency of testing and its solving time was significantly more than other methods. So we removed the strong branch in our experiment.
>
> | Method | Time             |
> | ------ | ---------------- |
> | SCIP   | 2.42 $\pm$ 11.25% |
> | GCNN       | 1.75 $\pm$ 9.73%     |
> | ML4CO-KIDA | 1.70 $\pm$ 9.60%     |
> | HRL-Aug    | **1.68** $\pm$ 9.44% |
> | Strong Branching | 17.11 $\pm$ 20.34% |
>
>
> >***Q4:*** "Offline DRL", which was initially discussed in some earlier papers, You might want to change the writing here to be more accurate.
>
> We really appreciate your valuable suggestion. The "offline DRL" is initially discussed in Fujimoto et al. (2018) [1], we will clarify this in the final version.
>
> >***Q5:*** Why there is no highlight on the best-accuracy method for each task?
>
> Upon reviewing the findings presented in Table 4, Table 5, and Table 6, it becomes evident that the accuracy of the different methods, namely our method, GCNN, and ML4CO-KIDA, does not correlate with their overall performance. Therefore, it is inappropriate to single out any specific method as the best-performing one.
>
> >***Q6:*** Why is it the case that the proposed method can reduce training time compared to other methods?
>
> As shown in Table 7 and Table 8, For the training time, the offline RL plays a significant role in reducing the training samples, the training sample size of our proposed HRL-Aug reduced by around 30% at each iteration compared with ML4CO-KIDA, which is the reason why our method can reduce training time compared to ML4CO-KIDA. Moreover, our online RL agents operate within a relatively small state-action space, involving only two actions. Hence, despite the general notion that RL agents can be time-consuming to train, our method effectively mitigates this concern by leveraging offline RL and operating within a compact state-action space, resulting in a negligible increase in training time compared to the overall training process.
>
> **References**:
>
> [1] Fujimoto S , Meger D , Precup D .Off-Policy Deep Reinforcement Learning without Exploration[J].  2018.

---

### Author Response · Authors · 2023-11-15
**The ordering of the Tables in the Experiments section is confusing.**

We appreciate your valuable suggestion; The non-sequential order of the tables was primarily driven by the constraints imposed by the paper's length limitations. To make the readers follow the descriptions effectively, we have reassigned the sorting of the tables in the updated PDF file.

---

### Author Response · Authors · 2023-11-21

Dear reviewers,

We want to express our sincere gratitude again for your valuable comments and thoughtful suggestions. During the author response period, we have undertaken every effort to address the concerns raised by the reviewers, our key responses are summarized as follows:

1.	Further discussion on the motivation behind our research, supported by relevant literature and empirical experimental results.
2.	We have expanded upon the novelty of our method, highlighting its unique contributions to the field.
3.	We have provided a detailed explanation of how both the online and offline agents function in our work.
4.	We have made enhancements to the overall presentation of our paper, ensuring clarity and coherence.

In our individual responses to each reviewer, we truly hope that our responses have met your expectations and assuaged any concerns. Since the discussion time window is very tight and is approaching its end, we genuinely do not want to miss the opportunity to engage in further discussions with you, which we hope could contribute to a more comprehensive evaluation of our work.

With heartfelt gratitude and warmest regards,

The Authors

---

### Meta-Review · Area_Chair_eKFC · 2023-12-09

**Metareview:**

This paper proposes a novel reinforcement learning framework for solving mixed integer programs (MIPs), by training a policy online to decide whether to use an expert policy or a learning-based policy to make decisions about the branching strategy. The paper compares their approach to several baselines, including prior learning-based approaches and the traditional open source solver SCIP, demonstrating that their approach outperforms them on several benchmarks, especially for harder problem instances. The authors also note that their approach outperforms Gurobi, a commercial solver (and likely the most widely used solver in practice), on one benchmark, though the improvements are relatively small.

The reviewers generally agreed that while the proposed framework is somewhat incremental, the results from the evaluation were compelling; integer programming is widely used and leveraging machine learning to improve state-of-the-art solvers is a compelling application. There were also some concerns about the complexity of the approach due to the need for online RL updates, though this complexity can be justified in settings where performance is critical. Finally, there were some concerns about the presentation that the authors have worked on addressing.

**Justification For Why Not Higher Score:**

The proposed approach is relatively incremental. Also, while the evaluation is compelling, Gurobi presumably still outperforms the authors' approach on most benchmarks (as far as I could tell, the authors only showed performance improvements results compared to Gurobi in Table 9).

**Justification For Why Not Lower Score:**

Integer linear programming is a compelling application of reinforcement learning, since it is a widely used tool and the results are directly transferrable to practice.

---

### Decision · Program_Chairs · 2024-01-16

Accept (poster)